# Rapid Deterioration and Fatal Outcomes in Colloid Cyst-Induced Obstructive Hydrocephalus: A Case Report

**DOI:** 10.3390/healthcare12212155

**Published:** 2024-10-29

**Authors:** Abdullah Basnawi, Alaa Alustath

**Affiliations:** 1Faculty of Medicine, University of Tabuk, Tabuk 47913, Saudi Arabia; 2International Medical Center Hospital, Jeddah 23214, Saudi Arabia; astath@imc.med.sa

**Keywords:** colloid cyst, hydrocephalus, obstructive hydrocephalus, intracranial pressure

## Abstract

**Introduction:** Colloid cysts are rare intracranial tumors that can cause obstructive hydrocephalus, a potentially life-threatening condition. Despite being typically benign, they often present with insidious symptoms, leading to delayed diagnosis and catastrophic outcomes. **Case Report:** A 29-year-old woman presented with a two-month history of worsening headaches, nausea, and vomiting. Neuroimaging revealed a colloid cyst obstructing the third ventricle, resulting in hydrocephalus. Despite emergency placement of an external ventricular drain, the patient’s neurological condition deteriorated rapidly, culminating in brain death. **Conclusions:** This case highlights the critical importance of the early diagnosis and aggressive management of colloid cyst-induced hydrocephalus. The rapid progression of symptoms and devastating outcomes underscore the need for increased awareness among healthcare providers. Given the high mortality associated with this condition, further research is warranted to identify predictive factors and develop effective treatment strategies.

## 1. Introduction

Colloid cysts, although uncommon, represent a distinct entity within the realm of intracranial neuroepithelial neoplasms. These gelatinous or lobulated lesions predominantly reside within the ventricular system, with the third ventricle being the most frequent location [1]. Their indolent growth pattern often belies their potential to disrupt the cerebrospinal fluid (CSF) dynamics, leading to a life-threatening condition known as obstructive hydrocephalus [2]. Obstructive hydrocephalus arises when an impediment hinders the normal flow of the CSF. Colloid cysts, strategically situated within the ventricular cavities, can effectively block the foramina of Monro or the aqueduct of Sylvius [3]. This obstruction triggers a cascade of events, culminating in progressive ventricular enlargement and a relentless rise in intracranial pressure (ICP). The clinical manifestations of this pressure surge encompass a triad of headache, nausea, and vomiting, often accompanied by gait ataxia, cognitive decline, and papilledema on funduscopic examination [4].

The clinical presentation of obstructive hydrocephalus can mimic other neurological ailments, potentially delaying diagnosis and intervention. Secondly, the surgical resection of the cyst itself carries inherent risks, such as hemorrhage, infection, and damage to surrounding neurovascular structures [5]. Moreover, the potential for recurrent hydrocephalus necessitates close post-operative monitoring and possible implementation of permanent CSF diversionary shunts [6]. The incidence of colloid cysts is estimated at about one per million individuals per year, constituting 0.5–1% of all intracranial tumors [7]. Notably, while colloid cysts are classified as benign tumors, their strategic location within the ventricular system can disrupt cerebrospinal fluid (CSF) flow, leading to life-threatening complications. In fact, colloid cysts account for 15–20% of all intraventricular masses [1,8]. Overall, colloid cysts pose a significant risk of neurological complications and have a symptomatic case mortality rate of 3.1% to 10% [9].

This case report delves into the intricate management course of a patient afflicted by colloidal cyst induced hydrocephalus.

## 2. Patient Information

A 29-year-old woman with a two-month history of recurrent headaches presented to the emergency department (ED) experiencing a significant exacerbation of her symptoms. The headaches, previously uninvestigated, had intensified in the past few days, now being accompanied by nausea, vomiting, and a marked decrease in appetite. Notably, the patient reported a recent air flight, after which her symptoms demonstrably worsened. Despite the concerning nature of her presentation, initial vital signs upon arrival were within normal limits. Her Glasgow Coma Scale (GCS) score was 15, indicating full consciousness and orientation.

## 3. Clinical Findings

A thorough clinical evaluation commenced in the ED. Intravenous analgesics, hydration fluids, and antiemetic medications were administered to address her immediate symptoms. To rule out potential underlying metabolic or infectious etiologies, a comprehensive laboratory workup was initiated, encompassing a complete blood count (CBC), renal function tests, and blood glucose level determinations. Fortunately, these investigations yielded normal results.

## 4. Timeline

The patient presented with a two-month history of recurrent headaches. Symptoms worsened in the days leading up to the emergency department visit, accompanied by nausea, vomiting, and decreased appetite. A recent air flight exacerbated symptoms.

## 5. Diagnostic Assessment

The decision to expedite a head CT scan was further reinforced by the sudden onset of seizure activity. The seizure rapidly progressed to status epilepticus, a life-threatening condition characterized by prolonged or recurrent seizures without regaining full consciousness. Coinciding with the seizure activity, the patient’s GCS score precipitously declined to 6, indicating a significant decline in mental status. This critical development necessitated immediate endotracheal intubation to secure her airway and establish respiratory support.

As shown in Figure 1 and Figure 2, neuroimaging with a head CT scan revealed a crucial finding: a colloid cyst situated at the roof of the third ventricle. This strategically located cyst was obstructing the normal flow of the cerebrospinal fluid (CSF), leading to obstructive hydrocephalus. Given the urgency of the situation, a prompt neurosurgical consultation was arranged.

## 6. Therapeutic Intervention

The neurosurgical team deemed immediate intervention necessary to address the hydrocephalus and alleviate the pressure on the brain. Consequently, an external ventricular drain (EVD) was placed. EVD served as a temporary conduit to drain excess CSF and reduce intracranial pressure (ICP). The procedure successfully stabilized the patient’s ICP, which had been significantly elevated, registering between 55 and 60 mmHg.

## 7. Follow-Up and Outcomes

Unfortunately, despite the timely intervention with EVD placement and initial ICP stabilization, the patient’s neurological condition did not demonstrate the anticipated improvement. Over the subsequent course, her neurological status continued to deteriorate, ultimately progressing to irreversible brain damage. After a comprehensive evaluation and two weeks of intensive care support, the patient was declared brain dead due to the devastating complications arising from colloid cyst-induced obstructive hydrocephalus.

## 8. Discussion

This case report presents a 29-year-old woman with a colloid cyst obstructing the third ventricle, leading to life-threatening obstructive hydrocephalus. Despite prompt intervention with external ventricular drainage (EVD) to alleviate intracranial pressure (ICP), the patient’s neurological condition deteriorated, culminating in irreversible brain damage and ultimately brain death. This case underscores the potential complexities associated with colloid cyst-induced hydrocephalus and the importance of early diagnosis and intervention to prevent devastating consequences.

The delay in diagnosis in this case may be attributed, in part, to the patient’s family’s socioeconomic status. Low-income families often face barriers to healthcare access, including financial constraints and cultural beliefs. This case highlights the importance of addressing these barriers to ensure the timely diagnosis and treatment of neurological conditions. As observed in our case, the patient’s initial presentation with recurrent headaches, which unfortunately remained uninvestigated for two months, might be a subtle manifestation of the developing hydrocephalus. The subsequent worsening of headaches, accompanied by nausea, vomiting, and the concerning detail of post-flight symptom exacerbation, points towards a possible increase in ICP due to the progressive CSF obstruction by the colloid cyst. Chronic headaches can be a presenting symptom of occult hydrocephalus, even in cases with normal or “normal pressure” ICP. A study by Edwards et al. [10] demonstrated this by reporting three patients with chronic headaches as the primary feature of idiopathic aqueduct stenosis, a cause of occult hydrocephalus. Notably, none of these patients had the classic early-morning headaches typically associated with increased ICP. However, all three experienced postural headaches that worsened in the head-down position. This, along with other suggestive symptoms like nausea and vomiting, prompted investigation with neuroimaging. Treatment with endoscopic third ventriculostomy, a procedure that bypasses the obstructed area and allows for CSF drainage, resulted in the complete resolution of headaches in all patients. This case series highlights the importance of considering occult hydrocephalus in the differential diagnosis of chronic headaches, particularly when accompanied by postural symptoms or other suggestive findings.

The sudden onset of seizure activity and rapid progression to status epilepticus further highlights the severity of the situation. While seizures are not an uncommon presentation in hydrocephalus, particularly when associated with rapid ICP rise [11,12], the life-threatening nature of status epilepticus underscores the critical need for urgent intervention. The neuroimaging findings, revealing a colloid cyst at the third ventricle causing obstructive hydrocephalus, aligned with the clinical presentation and mandated immediate action.

The placement of an EVD is a well-established strategy for managing acute hydrocephalus and reducing ICP [4]. In our case, the EVD successfully normalized the elevated ICP, offering a temporary reprieve. However, the lack of subsequent neurological improvement suggested that a more profound and potentially irreversible injury had already been inflicted upon the brain tissue. This delayed improvement is a recognized complication in severe hydrocephalus, particularly when intervention is delayed [13]. The delayed diagnosis and presentation with advanced hydrocephalus in our case might have contributed to the poor outcome.

Several studies have explored the management strategies and outcomes of colloid cyst-induced hydrocephalus. A retrospective analysis by Stachura et al. [14] reported successful outcomes in patients treated with microsurgical resection of the colloid cyst, highlighting the importance of definitive treatment alongside hydrocephalus management. The optimal surgical approach for colloid cysts in the third ventricle is debated, with complete removal traditionally preferred. However, a study by Zymberg et al. suggests that a more balanced approach might be beneficial. They reported that endoscopic cyst aspiration and coagulation, often leaving the capsule intact, achieved good outcomes. This approach resulted in symptom resolution, a low complication rate, and no recurrences. These findings suggest that while complete resection remains desirable in some cases, a more conservative strategy focused on symptom relief and minimizing procedural risks might be a viable alternative, particularly in limited-resource settings. Careful surgical planning and tailoring the resection strategy based on individual patient factors are crucial [15].

Our case tragically illustrates the potential for rapid neurological decline with colloid cysts. While benign, these cysts can obstruct cerebrospinal fluid flow, causing life-threatening hydrocephalus. A study by Singh et al. [9] identified risk factors and management strategies in such cases. They found that patients with hydrocephalus and a colloid cyst faced an extremely high risk of rapid decline and death. A larger cyst size was also associated with higher mortality. However, the most critical finding was the importance of prompt surgery. All patients without surgery died, whereas nearly half who received surgery survived. Interestingly, females with smaller cysts seemed more susceptible to rapid decline, suggesting that a potentially more aggressive treatment approach might be warranted in such cases. Singh et al.’s study highlights the importance of early diagnosis, prompt surgical intervention, and potentially using a more aggressive approach for females with acute neurological deterioration due to colloid cysts.

## 9. Conclusions

This case underscores the importance of early diagnosis and aggressive management in colloid cyst-induced hydrocephalus. While EVD placement can be a life-saving measure, careful monitoring and consideration of additional interventions, such as post-drainage CT scans and potential contralateral ventricle drainage, are crucial for optimizing outcomes. Early surgical intervention and addressing socioeconomic barriers to healthcare access to improve outcomes may also be considered in certain cases, depending on the patient’s condition and the risks involved.

## Figures and Tables

**Figure 1 healthcare-12-02155-f001:**
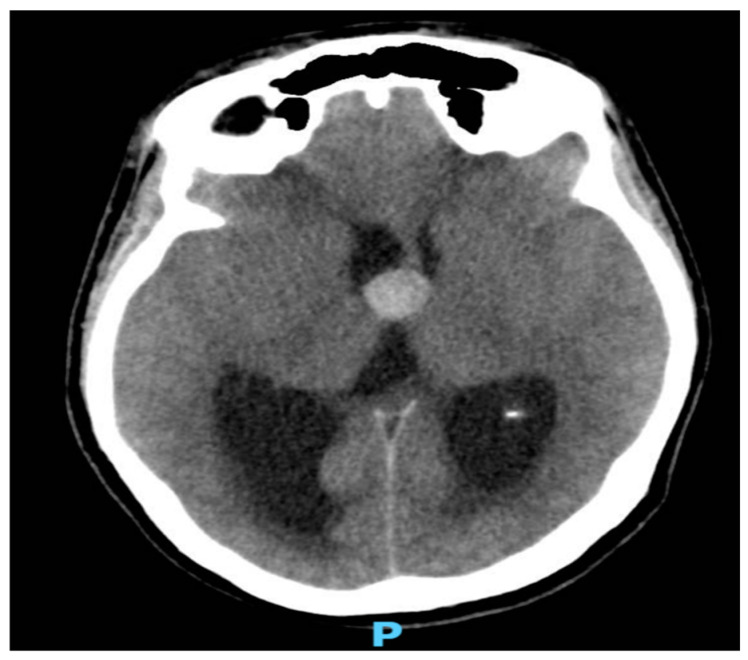
A nonenhanced CT image of the brain showing an oval dense lesion at the roof of third ventricle measuring 1.7 × 1.6 cm, obstructing the foramen of Monro.

**Figure 2 healthcare-12-02155-f002:**
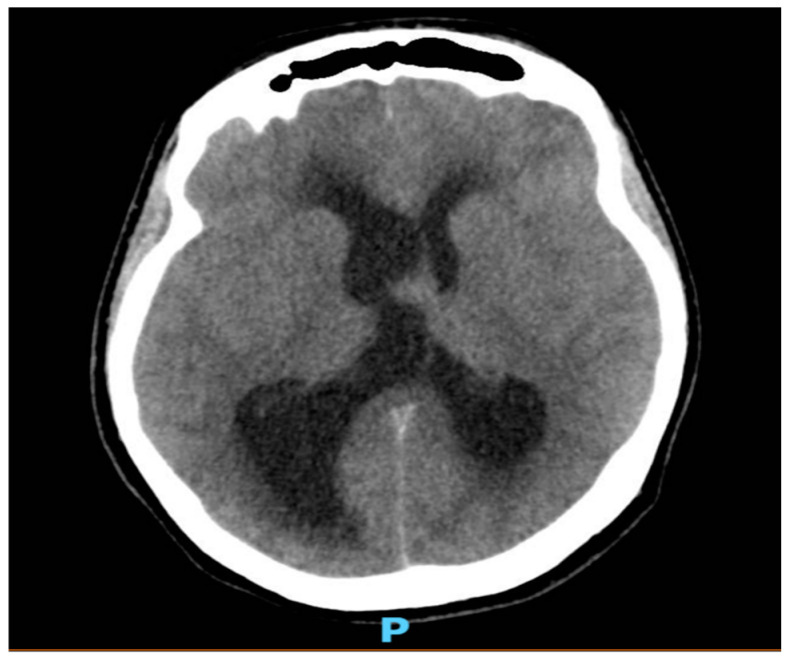
A nonenhanced CT image of the brain showing dilated lateral ventricles more on the right side with surrounded periventricular hypo-density in keeping with CSF permeation.

## Data Availability

Materials including a detailed patient timeline and additional neuroimaging studies, and the raw data supporting the conclusions of this case report will be made available by the authors on request.

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
