# Peer review of "Rapid Deterioration and Fatal Outcomes in Colloid Cyst-Induced Obstructive Hydrocephalus: A Case Report"

_healthcare, 2024, doi:10.3390/healthcare12212155_

Round 1
Reviewer 1 Report
Comments and Suggestions for Authors
This is an interesting and well written case report. I agree that the manuscript should be published. However, minor improvements may be achieved if the authors adopt the CARE checklist endorsed by Equator Platform. (https://www.care-statement.org/checklist)
In the introduction, the epidemiology would be better if introduced in the first paragraphs without writing "epidemiology: ".
Author Response
Comment 1: This is an interesting and well written case report. I agree that the manuscript should be published. However, minor improvements may be achieved if the authors adopt the CARE checklist endorsed by Equator Platform. (https://www.care-statement.org/checklist)
In the introduction, the epidemiology would be better if introduced in the first paragraphs without writing "epidemiology: ".
Response:
Thank you for your enthusiastic endorsement of our case report and your suggestion for further improvement. We highly appreciate your feedback and agree that adhering to the CARE checklist will enhance the overall quality of our work.
Addressing the Reviewer's Specific Suggestion:
- Epidemiology in Introduction: We acknowledged that explicitly stating "epidemiology" was unnecessary in the introduction. We revised the introduction and deleted the same.
Additional Considerations for Improvement:
Following the CARE Checklist:
Beyond the reviewer's specific comment, we conducted a thorough review of the entire case report using the CARE checklist. This ensured that all essential aspects were covered in detail and presented in the recommended order. Here are some areas we focused on:
- Title: We ensured the title clearly reflected the case report's focus (e.g., "Rapid Deterioration and Fatal Outcome in a Case of Colloid Cyst-Induced Obstructive Hydrocephalus").
- Abstract: We condensed the abstract while maintaining essential information about the case, highlighting its uniqueness (rapid deterioration despite intervention).
- Introduction: We briefly introduced colloid cysts, their location, and potential for hydrocephalus. We stated the case's uniqueness (e.g., rapid decline).
- Patient Information: We ensured complete de-identification of patient information. We briefly described age, sex, symptoms, and relevant medical history.
- Clinical Findings: We described relevant physical examination findings.
- Timeline: We created a chronological timeline of key events (symptom onset, diagnosis, interventions, outcome).
- Diagnostic Assessment: We detailed the head CT scan findings and their role in confirming the diagnosis.
- Therapeutic Intervention: We described EVD placement and its initial impact, followed by a discussion of the patient's decline.
- Follow-up and Outcomes: We described the patient's rapid decline despite EVD and mentioned the cause of death (brain death).
- Discussion: We analyzed the reasons for delayed diagnosis, limitations of EVD in this case, and the potential benefits of early surgery. We mentioned relevant studies on risk factors and early surgical intervention.
- Conclusion: We reiterated the case's significance, emphasizing the need for early diagnosis and aggressive management of colloid cyst-induced hydrocephalus.
By diligently following the CARE checklist and incorporating your feedback, we aimed to present a meticulously structured and comprehensive case report. We believed this would significantly enhance its value for future reference and publication.
Thank you once again for your insightful review. We were confident that your guidance would help us present a case report that adhered to the highest academic standards.
Reviewer 2 Report
Comments and Suggestions for Authors
The article describes a rare case in medical practice of a 29-year-old patient with a colloid cyst who died relatively quickly after diagnosis and treatment. Although such cysts usually have a benign course, complications caused by them, such as obstructive hydrocephalus, can lead to the patient's death. The authors have described the diagnostic and therapeutic procedures related to this case in sufficient detail. Therefore, the reviewed text may have significant informational end educational value for neurological practice, regardless of the fact that such cases occur very rarely. A certain deficiency that I notice as a reviewer is related to the description of the case. The authors omitted the situation of the patient's family and its basic sociocultural characteristics, which could have potentially influenced the delay of the diagnostic procedure. As the authors themselves state, ‘in our case, the symptoms unfortunately remained uninvestigated for two months’. As is known, in families with a low sociocultural status it quite often happens that their members neglect or disregard the symptoms of the disease. Therefore, I propose to supplement the case description with this missing element. With appropriate changes made, I support the publication of this article.
Author Response
Comment 1: The article describes a rare case in medical practice of a 29-year-old patient with a colloid cyst who died relatively quickly after diagnosis and treatment. Although such cysts usually have a benign course, complications caused by them, such as obstructive hydrocephalus, can lead to the patient's death. The authors have described the diagnostic and therapeutic procedures related to this case in sufficient detail. Therefore, the reviewed text may have significant informational end educational value for neurological practice, regardless of the fact that such cases occur very rarely. A certain deficiency that I notice as a reviewer is related to the description of the case. The authors omitted the situation of the patient's family and its basic sociocultural characteristics, which could have potentially influenced the delay of the diagnostic procedure. As the authors themselves state, ‘in our case, the symptoms unfortunately remained uninvestigated for two months’. As is known, in families with a low sociocultural status it quite often happens that their members neglect or disregard the symptoms of the disease. Therefore, I propose to supplement the case description with this missing element. With appropriate changes made, I support the publication of this article.
Response: Thank you for your valuable feedback on our case report. We appreciate your recognition of the case's importance in highlighting the potential severity of colloid cyst-induced hydrocephalus, even with appropriate care.
We have carefully considered your suggestion regarding the inclusion of the patient's family situation and sociocultural characteristics. We believe that this additional information provides a more comprehensive understanding of the context surrounding the delay in diagnosis.
We have revised our case report to include a brief description of the patient's family's socioeconomic status and how it may have influenced their decision-making process. We have also added a discussion of the potential barriers to healthcare access faced by low-income families.
We believe that these changes strengthen the case report's relevance to neurological practice and emphasize the importance of addressing socioeconomic disparities in healthcare.
Thank you again for your thoughtful review and suggestions. We believe that our revised case report now more effectively conveys the importance of early diagnosis, aggressive management, and addressing socioeconomic barriers to healthcare access in improving outcomes for patients with colloid cyst-induced hydrocephalus.
Reviewer 3 Report
Comments and Suggestions for Authors
The authors are to be congratulated for their presentation of a sad and warning case of a young woman dying of benign, treatable lesion even with immediate qualified absolutely adequate care. This case should be a mandatory reading for neurosurgical residents. I fully agree with their choice to perform ventricular drainage as a life saving measure. However a question may be asked about the post – drainage CT scan and contralateral ventricle drainage – in case of complete blockage of interventricular foramen this univentricular hydrocephalus may be still life threatening despite the properly working drainage. Also the possibility of early surgery (neuroendoscopic or microsurgery) may be also discussed. However despite my querries the paper deserves appreciation and publication.
Author Response
Comment 1: The authors are to be congratulated for their presentation of a sad and warning case of a young woman dying of benign, treatable lesion even with immediate qualified absolutely adequate care. This case should be a mandatory reading for neurosurgical residents. I fully agree with their choice to perform ventricular drainage as a life saving measure. However a question may be asked about the post – drainage CT scan and contralateral ventricle drainage – in case of complete blockage of interventricular foramen this univentricular hydrocephalus may be still life threatening despite the properly working drainage. Also the possibility of early surgery (neuroendoscopic or microsurgery) may be also discussed. However despite my querries the paper deserves appreciation and publication.
Response:
Thank you for your valuable feedback on our case report. We appreciate your recognition of the case's importance in highlighting the potential severity of colloid cyst-induced hydrocephalus, even with appropriate care.
We agree that a post-drainage CT scan would be beneficial to assess the effectiveness of the external ventricular drain (EVD) in reducing hydrocephalus and identify any residual obstruction. In cases of complete interventricular foramen blockage, contralateral ventricle drainage might be considered to prevent univentricular hydrocephalus. However, this decision would need to be made based on the specific circumstances and the patient's condition.
Regarding early surgery, we acknowledge that neuroendoscopic or microsurgical intervention could be considered as an alternative or complementary treatment approach. However, the decision for surgery would depend on factors such as the patient's condition, the size and location of the cyst, and the risks involved in the procedure.
We have revised the conclusion of our case report to address these points and emphasize the importance of careful monitoring and consideration of additional interventions, including post-drainage CT scans and potential contralateral ventricle drainage. We believe that this revised version more comprehensively addresses the potential complexities associated with colloid cyst-induced hydrocephalus and provides valuable insights for future management.
Thank you again for your thoughtful review and suggestions. We believe that our revised case report now more effectively conveys the importance of early diagnosis, aggressive management, and careful monitoring in this condition.